# Accelerating Augmentation Invariance Pretraining

**Jinhong Lin**[*]  **Cheng-En Wu**[*]  **Yibing Wei**  **Pedro Morgado**
University of Wisconsin–Madison
{jlin522, cwu356, wei96, pmorgado}@wisc.edu

## Abstract

Our work tackles the computational challenges of contrastive learning methods, particularly for the pretraining of Vision Transformers (ViTs). Despite the effectiveness of contrastive learning, the substantial computational resources required for training often hinder their practical application. To mitigate this issue, we propose an acceleration framework, leveraging ViT's unique ability to generalize across inputs of varying sequence lengths. Our method employs a mix of sequence compression strategies, including randomized token dropout and flexible patch scaling, to reduce the cost of gradient estimation and accelerate convergence. We further provide an in-depth analysis of the gradient estimation error of various acceleration strategies as well as their impact on downstream tasks, offering valuable insights into the trade-offs between acceleration and performance. We also propose a novel procedure to identify an optimal acceleration schedule to adjust the sequence compression ratios to the training progress, ensuring efficient training without sacrificing downstream performance. Our approach significantly reduces computational overhead across various self-supervised learning algorithms on large-scale datasets. In ImageNet, our method achieves speedups of $4\times$ in MoCo, $3.3\times$ in SimCLR, and $2.5\times$ in DINO, demonstrating substantial efficiency gains.

## 1   Introduction

Self-supervised learning (SSL) has emerged as a powerful pre-training paradigm, demonstrating remarkable success across a variety of domains. By designing pretext tasks that leverage unlabeled data, SSL eliminates the need for costly and labor-intensive manual annotations. Among SSL methods, contrastive [5, 16] and distillation-based [4, 24] learning are among the most effective. They learn representations through transformation invariance, meaning that the model learns to create similar representations for different augmentations of the same image, but different across images. These approaches have led to the development of state-of-the-art models for tasks ranging from image recognition [16, 12] to object detection [16] and video object segmentation [4].

Despite these achievements, SSL requires substantial computational resources for pretraining, with optimal performance often necessitating long training schedules, hindering their practical application. The computational demands of SSL are particularly high for Vision Transformers (ViTs), a promising class of neural networks that has recently gained significant attention for visual understanding tasks. ViTs represent images as sequences of patches, processed through self-attention layers. While self-attention enables the model to capture long-range dependencies and complex patterns more effectively than convolutional networks, their enhanced expressiveness (and weaker inductive bias) require even more extensive pre-training to achieve competitive performance. Our work aims to address these computational challenges by proposing an acceleration framework specifically tailored for ViTs.

Existing attempts to accelerate SSL pretraining primarily focus on defining improved learning rates and data augmentation schedules for faster learning [19] or increasing the strength of supervision through multiple targets [8]. While beneficial, these methods are not tailored to the ViT architecture, and can only provide limited acceleration. They also often change the underlying pretraining

---

[*]Equal Contribution

38th Conference on Neural Information Processing Systems (NeurIPS 2024).

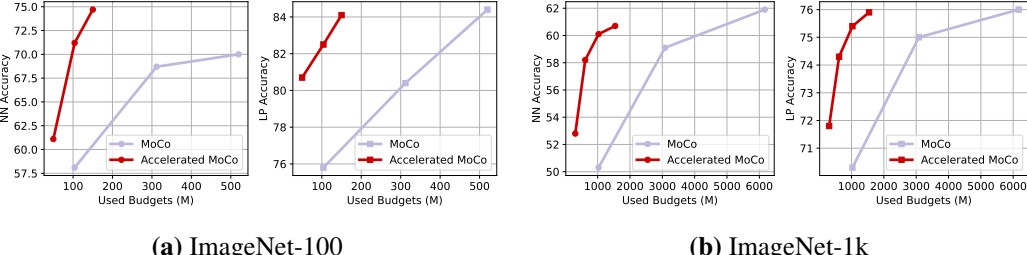

**(a)** ImageNet-100          **(b)** ImageNet-1k

**Figure 1:** Our accelerated MoCo-v3 achieves standard MoCo-v3 performance using only 1/5 of the training budget on ImageNet-100 and 1/3 on ImageNet-1k. The training budget (x-axis) is measured as the training time normalized by the forward pass of the base non-accelerated backbone model, in million (M) units. The results for ImageNet-100 are shown in Fig. 1a and for ImageNet-1k in Fig. 1b.

algorithm, making use of additional losses or data augmentations. Instead, we investigate acceleration techniques that leverage the ViT's unique ability to generalize across inputs of varying sequence lengths, while faithfully preserving the model's architectural design.

Since the time complexity of a training iteration is proportional to the input sequence length, our method identifies at each moment in time the most cost-effective mix of two simple sequence compression strategies: (1) randomized token dropout and (2) flexible patch scaling. We show that when applied judiciously with an appropriately optimized schedule, these simple strategies can significantly reduce the cost of gradient estimation, leading to faster convergence without compromising the quality of the learned representations (see Fig. 1). Our approach is general and can be applied to a wide range of SSL pre-training algorithms, as it only modifies the input sequence of the ViT. To demonstrate its effectiveness and generality, we apply our method for MoCo-V3 [16], SimCLR [5] and DINO [4], achieving significant training speed-ups on standard pre-training datasets like ImageNet-1K (between 2.5 to 4 times faster than the original methods). Additionally, we conduct a series of experiments on MoCo-V3 to perform a deeper and more general analysis of the proposed acceleration strategies. Through our analysis, we provide insights into the trade-offs between acceleration and performance, and the intricacies of establishing an optimal acceleration schedule. Specifically, we (1) investigate the gradient estimation error of various acceleration strategies alongside their performance on downstream tasks, study the impact of (2) compression rates on query and target sequences for contrastive learning, and (3) varying training budgets (showing that constant compression fails to meet peak model performance), and (4) establish an optimal acceleration schedule that adjusts to the training progress by minimizing the expected error of gradient estimation. Our analysis shows that, while the early phases of training typically benefit from aggressive acceleration strategies with high token dropout rates or large patch sizes, the gradient estimation biases increase as the model converges. Consequently, the optimal strategy should gradually shift towards smaller patches and lower dropout ratios.

## 2 Related Work

**Representation Learning Through Transformation Invariance** involves training models to produce consistent representations for augmented versions of the same image, primarily via contrastive learning and distillation methods. Contrastive learning achieves this by contrasting positive pairs, generated from augmenting the same image, with negative pairs from different images, as explored in numerous studies [13, 7, 6, 32, 23, 17, 1, 16, 5, 4]. Distillation methods [4, 24] focus on aligning embedding distributions across augmentations of varying scales without relying on negative samples. Both approaches effectively enhance feature quality without the need for labeled data.

**Accelerating Augmentation Invariance Pre-Training** Despite the high computational requirements, accelerating pre-training of vision transformers has remained underexplored. A few strategies have nevertheless been proposed. While focusing on ResNets, [19] introduced an architecture-agnostic method that dynamically adjusts augmentation intensity and learning rate schedules to hasten the training process. Some works focus on ViTs but modify its architecture or the underlying algorithm for acceleration. For example, [21] progressively merges tokens within the model, and [8] utilizes multiple small crops as positive pairs to increase the strength of supervision and promising convergence with smaller training budgets. In contrast, we propose a novel acceleration procedure for identifying the optimal schedule of simple sequence compression strategies, by ensuring that gradient estimation is cost-effective without introducing significant estimation biases.

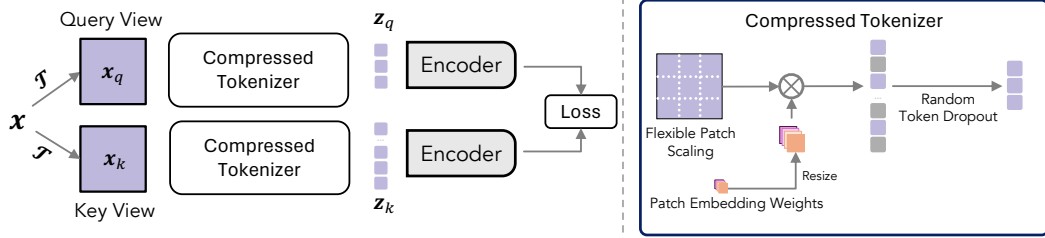

**Figure 2: Framework overview.** We propose a method for accelerating augmentation invariance pre-training of transformer neural networks. Acceleration is achieved by compressing the ViT's input sequence length using two strategies: (1) randomized token dropout and (2) flexible patch scaling. We further introduce a gradient error analysis framework to assess the efficacy of an acceleration strategy, enabling us to define an optimal acceleration schedule that adjusts to the training progress. The acceleration strategy can be applied to a variety of methods. For example, SimCLR optimizes both encoders by gradient descent, while MoCo and DINO use a momentum encoder to compute the representations for the Key view. The loss function also differs across algorithms.

**Accelerating Model Training** The computational demands of pretraining extend beyond contrastive learning. Various strategies have been proposed to mitigate this, such as random masking, a technique used in Vision Language pretraining to scale up training [22], and masked image modeling [14, 2] for in-context reconstruction. Other acceleration techniques include curriculum learning, starting with simple samples and progressively moving to harder ones [34, 26]; flexible ViT architectures that gradually increase in depth and width [25]; and resolution scaling, using low-resolution inputs at initial stages for faster convergence [20, 27, 30].

While these methods have been successful in accelerating pre-training, they often require changes to the model architecture or the underlying training algorithm. They have also been primarily designed for a variety of tasks beyond contrastive learning, and thus their findings and proposed algorithms may not be directly applicable. Our work covers this gap by proposing a novel acceleration framework tailored for augmentation invariance pre-training of ViTs. Furthermore, while the building blocks (token dropout and patch scaling) of our acceleration framework are closely related to existing methods, our work is the first to systematically analyze their impact on gradient estimation and to leverage this analysis to define an acceleration schedule that adjusts to the training progress.

## 3 [Background] Augmentation Invariance Pre-Training of ViTs

We present an approach to accelerate augmentation invariance pre-training of Vision Transformers. Since the proposed strategy only modifies the input sequence, it is applicable to a wide range of algorithms. For comprehensive empirical analysis, we apply our method to MoCo-V3 [7], SimCLR [5] and DINO [4].

### 3.1 Augmentation Invariance Pre-Training

While contrastive learning and distillation-based methods may differ in their implementation, they generally rely on augmentation invariance as the source of supervision. Given an input image $x$, two views are generated using a random data augmentation procedure $\mathcal{T}$ and often referred to as the query $x_q = \mathcal{T}(x)$ and the key $x_k = \mathcal{T}(x)$. These are processed by two encoders $f_q$ and $f_k$, producing query and key $n$-dimensional representations $z_q = f_q(x_q) \in \Re^n$ and $z_k = f_k(x_k) \in \Re^n$. Augmentation invariance is encouraged by aligning the query $z_q$ and key $z_k$ representations of the same image. Two major differences between algorithms are: the choice of the key encoder $f_k$ and the loss function leveraged to impose augmentation invariance.

**Key Encoder** While SimCLR uses a shared encoder to encode both views (*i.e.*, $f_k = f_q = f$), MoCo-V3 and DINO use a momentum encoder [28] for the keys $x_k$. Momentum encoders $f_k$ share the architecture, but their parameters are computed as the exponential moving average of the online encoder $f_q$. Momentum encoders have been shown to yield more stable target representations, and consequently improved representation learning.

**Loss function** Augmentation invariance can be enforced through a variety of loss functions. Among the chose algorithms, both SimCLR and MoCo-V3 leverage the InfoNCE loss [23]

$$\mathcal{L}_{\text{InfoNCE}} = -\log \frac{\exp\left(\text{sim}\left(z_q, z_k^+\right)/\tau\right)}{\exp\left(\text{sim}\left(z_q, z_k^+\right)/\tau\right) + \sum_{z_k^- \in \mathcal{Z}_k^-} \exp\left(\text{sim}\left(z_q, z_k^-\right)/\tau\right)}, \quad (1)$$

where $\texttt{sim}(\cdot, \cdot)$ is the cosine similarity, $\tau$ a temperature parameter, $\boldsymbol{z}_q$ and $\boldsymbol{z}_k^+$ the corresponding query and key representations, and $\mathcal{Z}_k^-$ a set of negative keys obtained from other images in the same batch. Unlike MoCo-V3 and SimCLR, DINO uses a distillation loss instead, which seeks to align query and key representations (also referred to as student and teacher's representations) without the explicit use of negatives samples. To accomplish this, query and key representations are first converted into a probability vector through a softmax operator $\boldsymbol{p} = SoftMax(\boldsymbol{z}/\tau) \in \Re^n$, and the model is trained to minimize the cross-entropy between the two

$$\mathcal{L}_{\text{dist}} = -\sum_{i=1}^{n} \boldsymbol{p}_{k,i} \log \boldsymbol{p}_{q,i}. \tag{2}$$

The sum is taken over the dimensions of the probability vector.

## 4 Gradient Acceleration Through Sequence Compression

While augmentation invariance can be used to learn representations with any type of neural network, the proposed acceleration procedure leverages properties specific to transformer models. In this work, we focused on Vision Transformers (ViTs) [10], a widely used architecture for vision tasks. ViTs process an image $x$ of resolution $H \times W$ by dividing it into a grid of $(H//p, W//p)$ patches, each of size $p \times p$. After embedding each patch into a $d$-dimensional vector and adding positional encodings to mark their spatial location, the ViT processes the sequence of patch embeddings through several self-attention transformer blocks [31]. Since transformer blocks' parameters are shared across the input sequence, the gradients of the loss wrt its parameters can be computed regardless of input sequence lenght. In this work, we tackle two key questions: (1) How to reduce the input sequence length with limited impact on the gradient estimation error? and (2) How to characterize the effectiveness of an acceleration strategy? We introduce two strategies, randomized token dropout and dynamic patch scaling, and propose a methodology to determine the effectiveness of gradient acceleration of a given strategy by analyzing its cost-adjusted bias-variance trade-off. This methodology allows us to identify the optimal acceleration strategy at each moment during training, eliminating the need for manual hyper-parameter tuning.

### 4.1 Randomized Token Dropout

Randomized token dropout (TknDrop) is a simple strategy for reducing the sequence length in ViTs, by simply removing a random subset of tokens from the input sequence. This strategy is especially effective in vision, since neighboring pixels are highly correlated and the model can still infer the visual content from a partial view of the image. However, while highly compressed sequences can speed up gradient estimation, too much compression may cause significant biases in the estimated gradients and consequently degraded model performance. Determining the optimal dropout rate is thus crucial for effective acceleration.

TknDrop is inspired by MIM methods [15, 2], which also mask the input sequence. However, while MIM uses masking to establish reconstruction targets for representation learning, we leverage token dropout to generate compressed input sequences to accelerate augmentation invariance pre-training.

### 4.2 Patch Scaling

The second strategy involves splitting the input image into a coarser grid of patches. As the sequence length $L$ is inversely proportional to the patch size $p$, larger patches allow us to reduce $L$ without removing any pixels from the input. However, since the patch embedding layer $W_{patch} : \Re^{p^2} \to \Re^n$ depends on a predefined patch size $p$, larger patches cannot be directly encoded. To mitigate this issue, we leverage the flexible patch embeddings introduced in [3], where $W_{patch}$ are dynamically resized to accommodate different patch sizes. Consider the weights $w_p \in \Re^{p^2}$ of a single output dimension of $W_{patch}$. Instead of simple interpolation, the optimal weights $w_q \in \Re^{q^2}$ at the larger size $q$ are computed by finding a projection $w_q = P w_p$ that minimizes the distance between the embedding of the original patch $x_p$ and the interpolated larger patch $x_{p \to q}$. Specifically, the optimal projection $P$ is obtained by solving

$$\arg\min_P \mathbb{E}_{x_p \in \mathcal{X}_p} \left[ (\langle x_p, w_p \rangle - \langle x_{p \to q}, P w_p \rangle)^2 \right], \tag{3}$$

where the expectation is taken over a distribution of patches $\mathcal{X}_p$.

| Method | Sample Cost ($C$) |
|---|---|
| SimCLR | $3\dfrac{L_q + L_k}{L_{base}}$ |
| MoCo-v3 | $\dfrac{3L_q + L_k}{L_{base}}$ |
| DINO | $\dfrac{3L_q + 3KL_q^{small} + L_k}{L_{base}}$ |

**Table 1:** Hardware-independent sample cost of different pre-training algorithms. We assume relatively short sequence lengths (typical of pre-training frameworks) where linear operations dominate over the quadratic self-attention operations.

| Patch \ Token Dropout Rate | 0.0 | 0.25 | 0.5 | 0.75 | 0.9 |
|---|---|---|---|---|---|
| 48 | 1.54 | 1.44 | 1.35 | 1.25 | 1.19 |
| 40 | 1.71 | 1.57 | 1.43 | 1.29 | 1.20 |
| 30 | 2.14 | 1.89 | 1.64 | 1.39 | 1.25 |
| 24 | 2.69 | 2.30 | 1.92 | 1.53 | 1.30 |
| 20 | 3.36 | 2.80 | 2.25 | 1.70 | 1.37 |
| 16 | 4.59 | 3.73 | 2.87 | 2.01 | 1.49 |

**Figure 3:** Accelerated MoCo-v3 sample costs for varying dropout rates and patch sizes. We assume uncompressed key sequences.

## 4.3 Combined Sequence Compression

Large patches can be trivially combined with token dropout by applying the two strategies in sequence. The sequence length are thus modulated by the selected patch size $q$ and token dropout rate $d$. Specifically, an image of size $H \times W$ split into a grid of $p \times p$ patches yields an uncompressed sequence of lenght $L = \left\lfloor \frac{HW}{p^2} \right\rfloor$. After compression, the sequence lenght is lowered to $L = (1-d) \times \left\lfloor \frac{HW}{q^2} \right\rfloor$.

## 4.4 Quantifying acceleration

**Linear complexity assumption.** Transformer blocks use two types of operations, token-wise transformations and self-attention. Token-wise operations, such as the MLP block or the query/key/value heads in self-attention, process each token in the sequence independently and thus scale linearly $\mathcal{O}(L)$ with its length $L$. Self-attention operations, on the other hand, establish relationships between all pairs of patches and thus scale quadratically $\mathcal{O}(L^2)$. However, the sequence length for most model pre-training frameworks is relatively small (tipically $L = 197$). Since there are many more linear operations than quadratic ones, the time complexity of linear operations dominates at this scale. Empirically, we observed that quadratic operations only become significant when the sequence length exceeds 400 patches. Thus, for the sake of simplicity, we assumed the time complexity of ViT pre-training to be linear with $L$.

**Sample costs of various algorithms.** Let the time spent per token be denoted as $t_{tkn}$. Then, under the linear complexity assumption, a forward pass takes approximately $t_{fwd} = L \times t_{tkn}$ seconds, and a backward pass twice as long, $t_{bwd} = 2L \times t_{tkn}$, as both the partial derivatives wrt the latent representations and the model parameters need to be computed [18]. Since, for SimCLR, the backward pass is performed on both encoders, the total time per sample is $t_{smp} = 3(L_q + L_k) \times t_{tkn}$. For MoCo-v3, the backward pass is only performed on the query encoder, and thus $t_{smp} = (3L_q + L_k) \times t_{tkn}$. DINO also uses $K$ smaller augmentations as additional query sequences for its distillation loss, further increasing the sample time to $t_{smp} = (3L_q + 3KL_q^{small} + L_k) \times t_{tkn}$. Finally, hardware dependencies (captured through $t_{tkn}$) can be removed by normalizing $t_{smp}$ by the forward pass of a standard input $t_{base} = L_{base} \times t_{tkn}$, where $L_{base} = 197$ is the sequence length for a regular $14 \times 14$ grid. Hardware independent sample costs are summarized in Table 1.

As can be seen, regardless of pre-training algorithm, the sample cost is proportional to the sequence lengths, $L_q$ and $L_k$. To visualize the impact of compression, we show the sample costs with varying token dropout ratios and patch sizes for MoCo-v3 in Fig. 3. These cost assume an uncompressed key sequence, as we empirically found that model pre-training is often more effective when supervision targets are computed without compression[2]. As can be seen, speed ups as large as $4\times$ can be achieved with 90% dropout rates and patches of size $q = 48$. However, usefull acceleration strategies should not only reduce the sample cost, but also minimize their impact on the estimated gradients.

## 5 Gradient Estimation Analysis of Acceleration Strategies

Given the large search space, empirically selecting the most effective strategy at each stage of training is computationally prohibitive. Instead, we posit that the distribution of the accelerated gradients should closely resemble that of the non-accelerated model. This criterion, further expanded below, can be used to inform us of the optimal mix of accelerated strategies at any point throughout training.

---

[2]Crops of size $240 \times 240$ were used, as this resolution is divisible by a larger set of patch sizes.

## 5.1 Formulation

**Gradient Distribution in Mini-Batch Training** Model optimization requires the minimization of a loss function $l(\boldsymbol{x}; \theta)$ wrt the model parameters $\theta$. Assuming independence between samples $\boldsymbol{x}$ in the training dataset $\mathcal{D}$, the expected loss is given by

$$\mathcal{L}(\theta) = \mathbb{E}_{\boldsymbol{x} \sim \mathcal{D}} [l(\boldsymbol{x}; \theta)] \approx \frac{1}{|\mathcal{B}|} \sum_{i \in \mathcal{B}} l(\boldsymbol{x}_i; \theta) \tag{4}$$

where $|\mathcal{B}|$ is the mini-batch size. Optimization algorithms update the model parameters $\theta$ using the gradient of the loss.

$$\nabla_\theta \mathcal{L}(\theta) = \mathbb{E}_{\boldsymbol{x} \sim \mathcal{D}} [\nabla_\theta l(\boldsymbol{x}; \theta)] \approx \frac{1}{|\mathcal{B}|} \sum_{i \in \mathcal{B}} \nabla_\theta l(\boldsymbol{x}_i; \theta) \tag{5}$$

For simplicity, denote the sample gradient as $g_\theta(\boldsymbol{x}) = \nabla_\theta l(\boldsymbol{x}_i; \theta)$, and the true gradient (computed over the entire dataset) as $G_\theta = \nabla_\theta \mathcal{L}(\theta)$. Since samples are independently drawn from the training dataset $\mathcal{D}$, then the sample gradient $g_\theta(\boldsymbol{x})$ is a random variable with mean and covariance given by

$$\mathbb{E}[g_\theta(\boldsymbol{x})] = G_\theta \quad \text{and} \quad \text{Cov}[g_\theta(\boldsymbol{x})] = \mathbb{E}\left[(g_\theta(\boldsymbol{x}) - G_\theta)(g_\theta(\boldsymbol{x}) - G_\theta)^T\right]. \tag{6}$$

Similarly, batch gradients are also unbiased estimates of the true gradient but with variance reduced by a factor of $|\mathcal{B}|$.

$$\mathbb{E}\left[\frac{1}{|\mathcal{B}|} \sum_{\boldsymbol{x} \in \mathcal{B}} g_\theta(\boldsymbol{x})\right] = G_\theta \quad \text{and} \quad \text{Cov}\left[\frac{1}{|\mathcal{B}|} \sum_{\boldsymbol{x} \in \mathcal{B}} g_\theta(\boldsymbol{x})\right] = \frac{1}{|\mathcal{B}|} \text{Cov}[g_\theta(\boldsymbol{x})]. \tag{7}$$

**Gradient Estimation Errors** When assessing an acceleration strategy, we need to consider both the mean and variance of the gradient estimates. While the bias is an intrinsic property of each strategy, the variance is a function of their computational cost. Strategies that reduce the computational cost significantly can potentially be used to average the gradients over a larger number of samples and thus reduce their variance. Thus, to fairly capture the bias-variance tradeoff of each strategy, we propose to use the *Mean Squared Error* (MSE) of the gradient estimate obtained with a *cost adjusted batch size*. From parameter estimation theory, it can be easily shown that the sample MSE decomposes into (squared) bias and variance components

$$\begin{aligned}
\text{MSE}\left(g_\theta^c(\boldsymbol{x}), G_\theta\right) & := \quad \mathbb{E}\left[\|g_\theta^c(\boldsymbol{x}) - G_\theta\|_2^2\right] \tag{8} \\
& = \quad \|G_\theta - \bar{g}_\theta^c\|_2^2 + \mathbb{E}\left[\|g_\theta^c(\boldsymbol{x}) - \bar{g}_\theta^c\|_2^2\right] = \text{Bias}^2(g_\theta^c, G_\theta) + \text{Var}(g_\theta^c) \tag{9}
\end{aligned}$$

where $\bar{g}_\theta^c = \mathbb{E}[g_\theta^c(\boldsymbol{x})]$ is the average accelerated gradient using strategy $c$. In other words, the MSE accounts for both the average deviation of the accelerated gradients from the ground truth and their variance across samples. Adjusting the MSE score for the cost of each strategy simply requires adjusting the variance by the number of samples within a fixed budget.

$$\text{CA-MSE}(g_\theta^c) = \text{Bias}^2(g_\theta^c, G_\theta) + \frac{\text{Cost}(c)}{\text{Budget}} \text{Var}(g_\theta^c). \tag{10}$$

**Estimating Bias and Variance** Both the bias and variance components can be efficiently computed given a model with parameters $\theta$ and a dataset $\mathcal{D}$. $G_\theta$ and $\bar{g}_\theta^c$ can be approximated by the sample average of the non-accelerated and accelerated gradients, respectively. While automatic differentiation libraries only track the aggregated gradient during batch processing, preventing us from directly computing the sample variance, we can estimate it by dividing the batch into smaller sub-batches (of size $K$) and adjusting the variance across sub-batches by $K$. Finally, it should be noted that to obtain a reliable estimate of CA-MSE, we need a large enough number of samples (we used 16k samples).

**Dynamic Acceleration** The cost-adjusted MSE provides a means to compare acceleration strategies without full training and evaluation. This metric could also be used to select the most effective strategy on-the-fly, at different stages during the training process. However, in practice, we conducted our analysis on intermediate checkpoints of a pre-trained model and used the findings to establish the "CA-MSE optimal" acceleration schedule for each method.

## 5.2 Analysis

In Section 4, we introduced two strategies to reduce the input sequence length (randomized token dropout and patch scaling) thereby reducing the cost of estimating the gradients of the model,

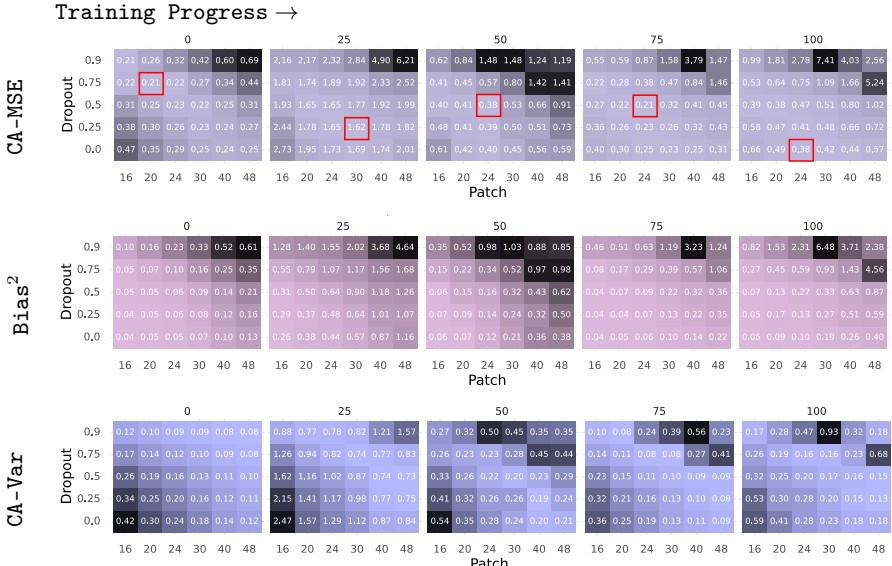

**Figure 4:** Error profile of accelerated gradients. From top to bottom, the three panels show the CA-MSE, squared bias and cost-adjusted variance of the gradient estimates, using different acceleration strategies and at different stages of training.

potentially at the expense of inaccurate gradients. To investigate their cost-accuracy trade-off, we measured the cost-adjusted MSE at 5 different stages of training progress, at 0%, 25%, 50%, 75% and 100% of training progress. We varied the dropout ratio in the set $\{0, 0.25, 0.5, 0.75, 0.9\}$ and the patch size in $\{16, 20, 24, 30, 40, 48\}$. Fig. 4 presents the CA-MSE score, normalized by the magnitude of the ground-truth gradient, across all configurations. Early in training (as seen in the first panel), both non-accelerated gradients (i.e., 0% dropout and patch size 16) and gradients derived from highly compressed input sequences (high dropout ratios and large patches) exhibit high MSE. However, non-accelerated gradients show low bias and high variance, while highly compressed gradients show high bias but low variance. More favorable trade-offs are achieved by combining moderate compression using both strategies simultaneously. As training advances and the model converges (remaining panels), the optimal strategy gradually shifts towards smaller patches and lower dropout ratios. This shift occurs because, as the model converges, gradient magnitudes shrink and the MSE becomes more sensitive to estimation biases.

# 6 Experiments

To assess the impact of acceleration on model performance, we conducted extensive exploration and ablation experiments, using the MoCo-V3 pre-training framework. We also assessed the generalizability of our methodology by accelerating other frameworks like DINO and SimCLR.

## 6.1 Experimental Setup

**Dataset** We conduct all experiments on the ImageNet dataset [9] using a ViT-Base transformer backbone for both the online and target encoders. Ablations and parametric studies are conducted on the ImageNet-100 (IN100) dataset, a randomly chosen subset of 100 classes from ImageNet. We adhere to the class partitioning used in previous studies [33, 29]. With around 125,000 images, IN100 provides a substantial amount of data for conducting statistically meaningful experiments. We also validate our findings on the full ImageNet-1k dataset to ensure the generalizability of our results.

**Downstream Evaluation** We assess the quality of the learned representations in three ways. In line with standard practices in self-supervised learning, we measure the classification accuracy on the pre-training dataset either using a linear probe (LP) with frozen features or after full model finetuning (FT). We also measure the nearest neighbor accuracy (NN) as an indicator of the effectiveness of the learned representations. All downstream evaluations are conducted without sequence compression.

**Pre-training settings** To ensure fair reproduction, we followed the official implementations. In the case of MoCo, the only modification was the use of a non-symmetric loss. Originally, the two augmentations $x_q$ and $x_k$ are used both as queries and targets, forming two pairs for each sample. However, this is equivalent to using only one pair, while doubling both the batch size and number

| Training Dataset | Batch Size | Epochs | Training Budget | NN | LP | FT |
|---|---|---|---|---|---|---|
| IN-100 | 512 | 200 | 104M | 58.1 | 75.8 | 88.0 |
| | | 600 | 312M | 68.7 | 80.4 | 90.4 |
| | | 1000 | 520M | **70.0** | **84.4** | **90.6** |
| IN-1k | 1024 | 100 | 1028M | 50.3 | 70.3 | 82.0 |
| | | 300 | 3084M | 59.1 | 75.0 | 82.3 |
| | | 600 | 6168M | **61.9** | **76.0** | **82.5** |
| IN-1k | 4096 | 300×2 | 6168M | – | 76.7 | 83.2 |

**Figure 5:** Non-accelerated MoCo-v3 across training budgets and datasets, and comparison to the publically released MoCo-V3 model (last row). The effective training epochs for the official MoCo implementation is doubled, as it uses a symmetric loss.

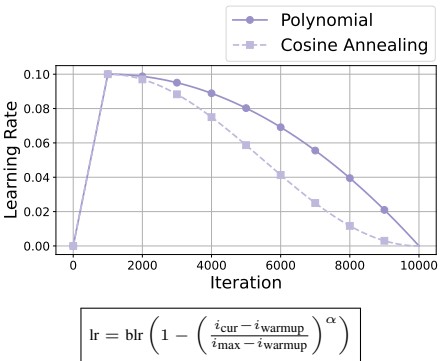

$$\text{lr} = \text{blr} \left( 1 - \left( \frac{i_{\text{cur}} - i_{\text{warmup}}}{i_{\text{max}} - i_{\text{warmup}}} \right)^{\alpha} \right)$$

**Figure 6:** Cosine vs Polynomial ($\alpha = 2$) learning rate decay schedules.

of epochs. The non-symmetric version also produces more diverse batches, which is advantageous given the use of batch normalization in the prediction heads.

To establish an optimized baseline on ImageNet-100, we empirically search for the learning rate, batch size and the required training budget (default values were used for other hyper-parameters). We observed that performance saturated for batch sizes of 512, and training budgets equivalent to 1000 epochs with a 40-epoch warmup phase. The optimal base learning rate was $5 \times 10^{-4}$, adjusted by the batch size scaling rule [11].

As for ImageNet-1k experiments, we followed the official training hyperparameters except for batch size, which was set to 1024 due to hardware limitations. MoCo's baseline performance on both ImageNet-100 and ImageNet-1K are shown in Fig. 5. As can be seen, despite the lower batch size, the model achieves comparable performance on ImageNet-1k (only 0.7% worse on both LP and FT accuracy).

**Accelerated MoCo Pre-training** As different gradient acceleration settings decrease the computational cost of a single iteration by different factors (see Fig. 3), fair comparisons require controlling the training budget, rather than the number of epochs. We express the training budget in units of the hardware independent sample costs defined in Section 4.4. On the ImageNet-100 dataset, where optimal performance was achieved at a training budget of 520M units (equivalent to 1000 epochs), we experiment with accelerated budgets ranging from 25M to 200M units. On ImageNet-1k, where baseline performance is achieved with a budget of 6168M, we varied the accelerated budget between 300M and 1500M.

Similarly to [19], we observed that the learning rate schedule also impacts the effectiveness of acceleration strategies. Augmentation invariance pretraining commonly employs a cosine decay schedule, which rapidly decreases in the second half of training. However, under constrained training budgets, this rapid decay hinders the model's ability to learn during the late stages. To address this, we use a polynomial decay schedule (see Fig. 6) to maintain a relatively higher learning rate in the later stages of training.

## 6.2 Constant Gradient Acceleration Strategies

We begin by assessing the representations obtained with each gradient acceleration strategy when applied uniformly and independently throughout training, using the MoCo-v3 pre-training framework.

**Randomized Token Dropout** We studied the impact of token dropout on the learning process. To assess the efficacy of this approach, we trained the model with varying dropout rates for the query and key sequences, adhering to a restricted training budget of 100M units (20% of the budget utilized for the optimal MoCo setup). The results detailed in Table 2a support three noteworthy observations. First, with randomized token dropout, it is beneficial to keep the key (target) sequence uncompressed to preserve maximum information when calculating the targets for the query (online) encoder. We refer to this strategy as asymmetric acceleration. Second, training MoCo-v3 without acceleration under the constrained training budget (as shown in the last row of Table 2a) yields significantly inferior results compared to any of the tested accelerated versions. For example, acceleration via asymmetric token dropout with $L_q = 50$ surpasses the non-accelerated model by 11.5% in NN accuracy and 4.0% in LP accuracy. This finding highlights the effectiveness of token dropout for accelerating the learning process. Finally, it is possible to compress the sequence too much, as evidenced by the performance degradation when using $L_q = 20$ (90% dropout rate). This result is

| $L_q$ | $L_k$ | **Cost** | **NN** | **LP** |
|---|---|---|---|---|
| 20 | 20 | 0.4 | 58.2 | 73.3 |
| | 40 | 0.5 | 59.2 | 74.9 |
| | 197 | 1.3 | **63.9** | **78.5** |
| 50 | 50 | 1.0 | 65.4 | 79.6 |
| | 100 | 1.3 | 67.0 | 78.7 |
| | 197 | 1.8 | **69.6** | **81.1** |
| 100 | 100 | 2.0 | 59.8 | 79.0 |
| | 197 | 2.5 | **69.3** | **81.4** |
| 197 | 197 | 4.0 | 58.1 | 77.1 |

**(a)** Sym and asym Tkn-Drop.

| $p$ | $L$ | **Cost** | **NN** | **LP** |
|---|---|---|---|---|
| 16 | 225 | 4.59 | 53.5 | 73.6 |
| 20 | 144 | 2.94 | 64.1 | 79.9 |
| 24 | 100 | 2.04 | 67.1 | **81.2** |
| 30 | 64 | 1.31 | **68.1** | 80.6 |
| 40 | 36 | 0.73 | 62.0 | 76.3 |

**(b)** Sym patch scaling.

| $p_q$ | $p_k$ | **Cost** | **NN** | **LP** |
|---|---|---|---|---|
| 30 | 16 | 2.1 | 45.3 | 71.7 |
| 30 | 20 | 1.7 | 53.8 | 77.3 |
| 30 | 24 | 1.5 | 64.3 | **80.6** |
| 30 | 30 | 1.3 | **68.1** | 80.5 |

**(c)** Asym patch scaling.

**Table 2:** Ablation studies of symmetric and asymmetric token dropout and patch scaling (training budget: $100M$).

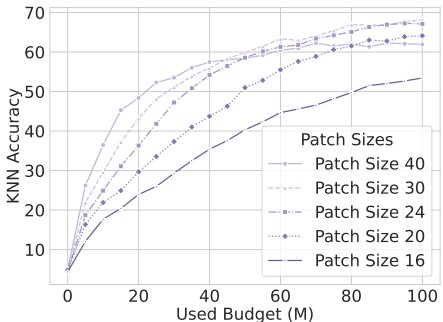

**Figure 7:** Training curves using constant symmetric patch scaling (training budget: $100M$).

consistent with the observations from the gradient error analysis (Fig. 4), which shows large gradient estimation biases for such high dropout rates.

**Patch Scaling** We also examined the impact of patch scaling on learned representations, maintaining a training budget of 100M units. We tested symmetric and asymmetric compression strategies for query and key sequences. For this, we used $240\times240$ resolution images to accommodate patches of size $16, 20, 24, 30$, and $40$, resulting in an uncompressed sequence length of 225 (with $p = 16$). The results, in Tables 2b and 2c, mirror the findings of token dropout. Models trained with larger patch sizes expedite learning, outperforming the non-accelerated model ($p = 16$) within the same budget, as shown in Fig. 7. Too much acceleration, with patches scaled above 30 pixels, can however degrade performance, which is aligned with the increased bias observed in the gradient error analysis. However, unlike token dropout, symmetric patch scaling, where both the query and key sequences are equally compressed, is more advantageous (see Table 2c). This is likely because patch scaling modifies the distribution of the input patches, making it preferable to maintain the same distribution for both the query and key sequences.

**Accelerated Training Across Training Budgets** The previous experiments showcased the efficacy of the proposed acceleration strategies within a constrained budget. To characterize their performance across an array of training budgets, we employed two representative strategies: asymmetric token dropout with $L_q = 50$ and $L_k = 197$, and symmetric patch scaling with $p = 30$. We trained the model with increasing budgets, from 25M to 200M units, and evaluated their downstream performance. The results, shown in Table 3, unveil a notable limitation of constant acceleration strategies. While these strategies are effective at lower budgets, they can overfit when the budget is increased. This is especially evident in the case of token dropout, as shown in Fig. 8. As a result of this overfitting, although the proposed acceleration strategies can outperform the non-accelerated model at lower budgets, they fail to meet their peak performance. As suggested by the gradient error analysis in Fig. 4, this overfitting can be traced back to the increased biases of the accelerated gradients often witnessed in the final stages of training. This occurs because, as the model converges, the gradient strength diminishes, allowing biases introduced by the compressed sequences to exert a greater influence on the learning process.

### 6.3 Optimized Acceleration Schedules

To circumvent the overfitting issue, we investigate optimized acceleration schedules that favor higher acceleration at the beginning and lower acceleration at the end of training. Although we can manually specify a schedule based on reasonable intuitions, the defined schedule would likely be suboptimal. Instead, we automate this process by leveraging the gradient error analysis of Fig. 4 to establish a "CA-MSE optimal" acceleration schedule.

As expected, at the beginning of training, a high token drop ratio and larger patch sizes have lower cost-adjusted MSE. However, as the training progresses, smaller patch sizes and lower drop ratios become more effective. We used these automatically derived schedules to train a model with three training budgets: 50M, 104M and 150M. As shown by the training curves in Fig. 8, by lowering the

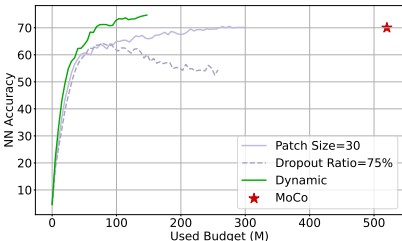

**Figure 8:** Training curve of three acceleration strategies: constant patch size, constant token dropout, and dynamic scheduling of joint patch scaling and token dropout.

| Training Budget | TknDrop | | PatchScale | |
|---|---|---|---|---|
| | NN | LP | NN | LP |
| 25M | 37.3 | 64.7 | 37.4 | 66.5 |
| 50M | 56.6 | 76.0 | 50.1 | 74.7 |
| 75M | 65.3 | 80.2 | 61.1 | 79.4 |
| 100M | **69.6** | **81.2** | 66.7 | 80.5 |
| 150M | 68.2 | 80.8 | 68.9 | 82.1 |
| 200M | 65.3 | 80.6 | **70.1** | **83.2** |

**Table 3:** Impact of training budgets on gradient acceleration strategies. Asymmetric token dropout ($L_q = 50$, $L_k = 197$) and symmetric patch scaling ($p = 30$).

acceleration towards the end of training, the model no longer overfits and is capable of reproducing the peak performance of the MoCo-v3 baseline in less than 30% of the time.

### 6.4 Accelerating Pretraining on ImageNet-1k

Finally, to validate the generalizability of our findings, we deploy the proposed optimized acceleration scheduled on the full ImageNet-1k dataset. Due to the unique characteristics of different SSL algorithms, we tailor our method slightly for each algorithm. As observed in 6.2, token dropout should be applied only to the online encoder, while patch scaling should remain consistent across encoders. However, since SimCLR directly optimizes over both encoders, treating them as online models, we found that its better to apply token dropout to both sequences for SimCLR. As for DINO, we empir-

| Algorithm | Accel. | Budget (M) | NN | LP | FT |
|---|---|---|---|---|---|
| MoCo | ✓ | 1542 | 60.7 | 75.9 | 81.9 |
| | | 6168 | 61.9 | 76.0 | 81.8 |
| SimCLR | ✓ | 922 | 50.7 | 68.4 | 81.5 |
| | | 3075 | 50.2 | 68.3 | 81.3 |
| DINO | ✓ | 1138 | 66.0 | 77.4 | 82.0 |
| | | 2846 | 67.3 | 77.4 | 81.8 |

**Table 4:** Acceleration of three augmentation invariance pretraining algorithms on ImageNet-1K. "Accel" indicated the use of the optimized acceleration schedule.

ically found that the additional small crops used as queries are already compressed enough, and applying additional compression to these sequences is not beneficial.

The results, shown in Table 4, demonstrate that the dynamic acceleration strategy is capable of achieving competitive performance with the non-accelerated model, while significantly reducing the computational requirements for training. For instance, our method achieves comparable LP accuracy for MoCo-V3 (75.9% vs 76.0%) with only 25% of the original budget (1542M vs 6168M iterations). We also achieved significant speedups for other pre-training frameworks, namely 2.5x for DINO and 3.3x for SimCLR.

## 7 Conclusion

In this paper, we propose a general acceleration framework for self-supervised learning that leverages simple sequence compression strategies to reduce the cost of gradient estimation. Our method is shown to significantly speed up the convergence of a variety of self-supervised methods, including constrastive (MoCo, SimCLR), and distillation-based frameworks (DINO) thus demonstrating its broad usability.

Given the compute-intensive nature of model pre-training and its implications on reproducibility, energy costs, and carbon footprints, we believe that further research on accelerated training is essential for advancing sustainable AI practices. Our paper aims to inspire continued exploration in this area, promoting the development of more efficient training methodologies that can (1) reduce the environmental impact of machine learning and (2) improve accessibility for researchers with limited resources.

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
