# OpenReview forum: "Accelerating Augmentation Invariance Pretraining"
_NeurIPS.cc/2024/Conference — NeurIPS 2024 poster_

### Official Review · Reviewer_95eS · 2024-06-19

**Soundness:** 2
**Presentation:** 2
**Contribution:** 2
**Rating:** 3
**Confidence:** 2

**Summary:**

This submission proposed to accelerate the training of ViT by two methods: 1) Randomly drop tokens for input. 2) Dynamically resize patches into different dimension. The second method is published in previous works.

**Strengths:**

N.A.

**Weaknesses:**

1. The novelity of the sumission is limited: In first propsoed method, dropping tokens randomly (masking) is a common trick for performance improvement (as cited by authors). It is natural to connect improvement to training acceleration. For the second method, as author point out that the method comes from published work.

**Questions:**

1. In Sec. 4.2, author mentioned that "large patches cannot be directly encoded", why large patches cannot be encoded (to what and by what?) Please add more explanation on the motivation of patch scaling.

**Limitations:**

N.A.

---

> ### Author Rebuttal · Authors · 2024-08-07
>
> We are sad that the reviewer completely misunderstood the contributions and the significance of the proposed work. We made clear both in the related work and method sections that Token Dropout and Patch Scaling are NOT contributions of our work, by explicitly citing the origins of each technique. The contribution of our work is how to design an optimized acceleration schedule, based on gradient estimation error analysis, so as to optimally define how much compression to use at different phases of training, and how to optimally combine these previously proposed techniques to achieve the required compression rate. We urge the reviewer to read the paper again and carefully reassess it based on our actual contributions.  We will be responsive during the author-reviewer discussion phase, in case there are any concerns we can address.
> ## Why large patches cannot be encoded (to what and by what?)
> The first step in a VIT is to embed each $p \times p$ patch into a fixed-size embedding vector (of size $n$). This is done by a linear projection matrix of size, $p^2 \times n$ (or equivalently implemented as a conv layer of kernel size $p$). Since the projection matrix is fixed size, it can’t be applied to larger (or smaller) patches.

---

> > ### Author Response · Authors · 2024-08-10
> > **Discussion**
> >
> > Dear Reviewer,
> >
> > Thank you for your thoughtful review and the time you’ve invested in evaluating our work. We have carefully addressed the points you raised in our rebuttal, and we would greatly appreciate the opportunity to clarify or discuss any remaining questions or concerns you may have.
> >
> > Thank you once again for your valuable feedback.

---

> > > ### Author Response · Authors · 2024-08-13
> > > **Final comment**
> > >
> > > Please read our final message to all reviewers. We truly appreciate your efforts in reviewing our paper. We hope you consider upgrading the final score to reflect the significant improvements made to the paper during the review cycle, and hopefully a renewed understanding of the intended contributions of our work.

---

### Official Review · Reviewer_Q1ps · 2024-07-03

**Soundness:** 3
**Presentation:** 3
**Contribution:** 2
**Rating:** 5
**Confidence:** 4

**Summary:**

This paper presents an acceleration framework for Vision Transformers in contrastive learning. It utilizerandomized token dropout and patch-scaling to reduce the sequence length to accelerate training. Based on an analysis of the gradient estimation error, this paper proposes an automated procedure to identify an optimal acceleration schedule. Extensive experiments demonstrate that accelerated pretraining achieves comparable performance on visual understanding tasks, while effectively reducing computation costs.

**Strengths:**

Strength：

1. The motivation is clear and reasonable.
2. Extensive experiments demonstrate the approach's effectiveness in reducing the training time while keeping comparable comparisons.
3. This paper is well-organized and clearly-written.

**Weaknesses:**

Weakness:

1. Lower ceiling. It seems like that this framework will lead to a decrease in the optimal performance of pretraining. For  instance, in Fig.1(b), Accelerated MoCo's highest performance is lower than MoCo, although much faster. By the way, ImageNet-1K performance is much more important than ImageNet-100, especially for a SSL method.
2. Missing important metric. This paper reports NN and Linear Probing to evaluate the approach, however, full-finetuning top-1 accuracy (on ImageNet-1K) is a critical metric for SSL method. Adding this metric can make this paper more convincing.
3. Changed structure. Since this paper utilizes flexible patch embedding, the structure is not a naive ViT anymore, which may undermine the application of this paper in many scenarios.
4. Limited evaluation. This paper only conduct experiments based on MoCo-v3, more evaluation on other SSL frameworks (e.g, another contrastive learning method or a MIM method) will make it solid.

**Questions:**

Please see the comments above.

**Limitations:**

Please see the comments above.

---

> ### Author Rebuttal · Authors · 2024-08-07
>
> We appreciate the valuable feedback. Below we answer the main concerns, and we will revise the paper accordingly. If there are any remaining concerns we can clarify or provide additional results/analysis, please let us know. We will be responsive during the author-reviewer discussion phase. Given the added experiments and evaluations, as well as the importance of accelerating contrastive pretraining, we hope the reviewer will reconsider the recommendation score.
> ## Lower ceiling
> First, we note that evaluating SSL methods is not a trivial task, and different researchers use/advocate for different evaluation metrics. Having said that, the most commonly used metric for assessing representation learning of SSL methods is Linear Probing accuracy on ImageNet-1k. Under this metric, the acceleration procedure achieves the **same** performance (only 0.1% lower) to the baseline MoCo. We understand that the NN accuracy is slightly worse (1.2% lower). However, this metric is known to be more sensitive to slight changes in the model since no weights are trained to adapt the model output to the downstream task. While we find it informative (the reason why we added it to the paper), linear probing is a more important metric. Also, while we understand that ImageNet-1K results are more important than ImageNet-100, the significantly higher NN accuracy in ImageNet-100 is still worth pointing out.
> ## Full finetuning
> Thanks for the suggestion. We plan to add full finetuning results to the paper. Unfortunately, some of the checkpoints we had trained were lost. We are in the process of retraining and will provide the results, hopefully, during the author-reviewer discussion phase.
> ## Changed VIT architecture
> We believe there was a misunderstanding. Patch scaling is only used for accelerating training. At inference time, the learned VIT simply reverts to the original patches of size 16. No changes were made to the model’s inference computational graph.
> ## Evaluation beyond MoCo
> This is a great suggestion. We have indeed been working on creating optimized acceleration schedules for algorithms beyond MoCo, using the technique proposed in the paper. We were able to achieve a 2.5x speedup in DINO pre-training and a 3.3x speedup in SimCLR pre-training.
>
>
> | Algorithm | Acceleration | Training Budget (M) | NN (%) | LP (%) | FT (%) |
> |-----------|--------------|---------------------|--------------|----------|----------|
> | **SimCLR** | ✔            | 922                 | 50.70        | 68.43    | 81.55    |
> |           | ✗            | 3075                | 50.22        | 68.33    | 81.39    |
> | **DINO (4 small crops)** | ✔            | 1138                | 66.00        | 77.42    | 82.01    |
> |           | ✗            | 2846                | 67.36        | 77.48    | 81.87    |
>
> where NN, LP, and FT refer to the accuracies of near neighbor, linear probing, and fine-tuning, respectively. For SimCLR, we simply replaced the backbone in the original implementation (from a ConvNet to ViT-base). As for DINO, the original method performs contrastive learning on both large and small crops. The small crops have a lower computational burden and thus can have a similar effect of speeding up training. To provide a realistic comparison to DINO, both the baseline (unaccelerated) and our accelerated version still use small crops (4 to be exact) in addition to the 2 large crops, with acceleration only applied to large crops. We will add to the paper the results above as well as an analysis of the impact of varying numbers of small crops in DINO

---

> > ### Author Response · Authors · 2024-08-10
> > **Discussion**
> >
> > Dear Reviewer,
> >
> > Thank you for your thoughtful review and the time you’ve invested in evaluating our work. We have carefully addressed the points you raised in our rebuttal, and we would greatly appreciate the opportunity to clarify or discuss any remaining questions or concerns you may have.
> >
> > Thank you once again for your valuable feedback.

---

> ### Author Response · Authors · 2024-08-13
> **Final comment**
>
> Dear reviewer,
>
> As promised, we evaluated the accelerated models trained with different training budgets using the full finetuning protocol and compared them to our baseline MoCo model. (We followed the MoCo-V3 paper and code in our finetuning evaluations). As can be seen in the table below, under this evaluation protocol, the speedup is even more pronounced (compared to what we showed in the paper using Linear Probe evaluations). Specifically, the model trained with about 10% of the budget already achieved a result comparable to unaccelerated training (only 0.24% worse).
>
> Overall, the main concerns were 1) the lower ceiling in some evaluations, 2) the absence of finetuning results, 3) the changed backbone architecture, and 4) the lack of experiments beyond MoCo. Additional experiments for 2 and 4 were conducted and will be added to the paper. We believe that concern #3 was a misunderstanding and concern #1 should now be less critical, given that we observe similar performances on both linear probing and finetuning (of all conducted experiments, only nearest neighbor evaluations on IN1k showed slightly lower performance ceilings). We thank the reviewer for helping us improve our paper, and ask the reviewer to reassess the final score to account for these improvements.
>
> | Algorithm | Acceleration | Training Budget (M) | FT (%) |
> |:----:|:----:|:-----:|:----:|
> | **MoCo** | ✔| 308| 80.56|
> |**MoCo** | ✔ | 617 | 81.61 |
> | **MoCo**| ✔ | 1080 | 81.81|
> | **MoCo**| ✔ | 1542 | 81.92 |
> |**MoCo** | ✗ | 6150 | 81.85 * |
>
> \* As mentioned in the paper, our non-accelerated model is slightly worse than the officially released MoCo-v3 model. While we used the same overall training budget, we could only train with relatively smaller batch sizes of 1024 (as opposed to the original 4096), which to the best of our knowledge is the cause of the gap. However, keep in mind that, the goal of the paper is to propose and validate a training acceleration technique. The results above show that, given the same training loss (which in the case of MoCo depends on the batch size), our acceleration method can significantly speed up convergence. We expect similar convergence speedups when training with higher batch sizes.

---

### Official Review · Reviewer_geFN · 2024-07-13

**Soundness:** 3
**Presentation:** 4
**Contribution:** 2
**Rating:** 6
**Confidence:** 4

**Summary:**

This work focuses on speeding up contrastive learning with vision transformers. Two methods, specifically tailored to ViTs, are investigated for making pretraining more efficient: randomised token dropout and flexible patch scaling. Additionally, the authors analyse the gradient estimation errors from these methods and create an automated strategy for optimal acceleration during pretraining. The resulting approach can achieve similar performances for a fraction of the budget, specifically 1/5 for ImageNet-100 and 1/3 for ImageNet-1k.

**Strengths:**

**Originality**
TknDrop is clearly inspired by common usage in MIM and patch scaling is taken from FlexiViT. These are therefore not original ideas but have not been explored for speeding up contrastive ViTs. The more original aspect of the paper, however, is the automatic scheduling of these techniques through gradient error monitoring.

**Quality**
The construction of the final dynamic acceleration method is methodical and thorough. Extensive ablations are performed to find the optimal combination that achieves the most effective acceleration. The experimental section overall is of very high quality.

**Clarity**
The paper is very clearly written throughout, and the structure is very natural and easy to follow. The plentiful figures and tables are effectively communicating the right information, though Figure 7 could have a larger font. It is however unclear what Lq=50 means in terms of the token dropout ratios (0, 0.25, 0.5, 0.75, 0.9). Does it mean that 50 tokens are kept out of the total 197 (meaning roughly 0.75 for the dropout ratio)?

**Significance**
The simplicity and automated nature of the method makes it seem easy for practitioners and researchers to use themselves. This can indeed facilitate a wider adoption of self-supervised pretraining for those with modest compute resources. However, while the authors point out that their narrow focus on MoCov3 + contrastive learning + ViT allows for a deeper study, the paper suffers a bit by not showing any generalisation to other types of training, like supervised pre-training on ImageNet.

**Weaknesses:**

My main concern is that the scope of the paper is rather narrow. It does one thing and it does it well, but this limits its potential impact. The proposed method is not limited to only contrastive pretraining and a single experiment that shows how it also applies to supervised pretraining on ImageNet would help show how generally applicable it is.

It’s not quite clear how the values 1/3 and 1/5 for the budget are obtained for the caption of Figure 1. Section 6.4 claims a 4x speedup on ImageNet. These claims can be made more consistent throughout.

**Questions:**

Are the speedup techniques explored in this paper orthogonal/complementary to e.g. resolution scaling or curriculum learning that have been explored in other works?

Can longer training with token dropout and patch scaling yield better results than the baseline, I.e. using the same budget of 520M on ImagetNet-100?

Are the overheads for automatically scheduling the acceleration accounted for in the budget, or are the optimal decision points for the scheduler computed offline and fixed before training?

**Limitations:**

There is no section that discusses the limitations of the proposed method explicitly. I would like to see some of the questions I’ve asked answered in such a section, or as part of the conclusion.

---

> ### Author Rebuttal · Authors · 2024-08-07
>
> We appreciate the valuable feedback. We are glad to see the originality of the proposed method and the significance and quality of our empirical results appreciated. We will revise the paper to add experiments and clarify unclear points, as described below. If there are any remaining concerns we can clarify, or provide additional results/analysis, please let us know. We will be responsive during the author-reviewer discussion phase.
> ## Generalization to other pretraining frameworks
> While we plan to investigate other pretraining frameworks such as supervised learning, vision-language pretraining, and even object detection/segmentation in a future journal extension of the paper, we focused on contrastive pre-training since contrastive learning is famously slow to converge, often requiring up to 1000 epochs to obtain optimal performance. Having said that, we have indeed been working on creating optimized acceleration schedules for algorithms beyond MoCo, using the technique proposed in the paper. We were able to achieve a 2.5x speedup in DINO pre-training and a 3.3x speedup in SimCLR pre-training.
>
> | Algorithm | Acceleration | Training Budget (M) | NN (%) | LP (%) | FT (%) |
> |-----------|--------------|---------------------|--------------|----------|----------|
> | **SimCLR** | ✔            | 922                 | 50.70        | 68.43    | 81.55    |
> |           | ✗            | 3075                | 50.22        | 68.33    | 81.39    |
> | **DINO (4 small crops)** | ✔            | 1138                | 66.00        | 77.42    | 82.01    |
> |           | ✗            | 2846                | 67.36        | 77.48    | 81.87    |
>
> where NN, LP, and FT refer to the accuracies of near neighbor, linear probing, and fine-tuning, respectively. For SimCLR, we simply replaced the backbone in the original implementation (from a ConvNet to ViT-base). As for DINO, the original method performs contrastive learning on both large and small crops. The small crops have a lower computational burden and thus can have a similar effect of speeding up training. To provide a realistic comparison to DINO, both the baseline (unaccelerated) and our accelerated version still use small crops (4 to be exact) in addition to the 2 large crops, with acceleration only applied to large crops. We will add to the paper the results above as well as an analysis of the impact of varying numbers of small crops in DINO.
>
>
> ## Relation to other speedup techniques
> The main difference between the prior techniques mentioned in the paper and our work is that prior works are not tailored to VITs. Curriculum learning strategies can be applied to any model and used in conjunction with our VIT-specific techniques for potentially larger speedups.
> Resolution scaling explores a similar idea to dynamic patch scaling. The main difference is that resolution scaling simply resizes the input images, while patch scaling adjusts the patch projection layer instead, in a more principled fashion (as extensively discussed in FlexiVIT). Prior work has explored resolution scaling mostly with CNNs, but it could also be extended to VITs. Given the similarities, combining resolution and patch scaling should not lead to major improvements.
> ## Longer training schedule
> In the algorithms that we tested, longer training schedules did not lead to improved performance. This is likely because we have chosen algorithms that have been fully optimized until convergence. As can be seen in Fig 7 of the main paper, when acceleration is used for too long, the model can overfit (NN accuracy of the model trained with a constant 75% dropout ratio drops in the 2nd half of training). We found that using long schedules will often lead to the overuse of acceleration and consequently performance drops. These drops can be recovered later on, after the schedule prescribes less acceleration, but still, equal or lower performance was observed at the end of training.
> ## How is the scheduler computed?
> The schedule is computed offline and fixed before training. The same optimized schedule is then used across many runs in the paper (eg, with different total budgets). Since the schedule is fixed, it has no additional overhead during training.
> ## Discussion of limitations
> Thank you for pointing this out. We will add a discussion of limitations/future work to the conclusion. In particular, we will highlight the potential of the method being used for other pre-training frameworks, as well as, the potential of the proposed method to be deployed in an online fashion for truly dynamic acceleration schedules.
> ## What Lq=50 means?
> It means that a total sequence length of 50 tokens (out of the initial 196) is fed to the query encoder.

---

> > ### Author Response · Authors · 2024-08-10
> > **Discussion**
> >
> > Dear Reviewer,
> >
> > Thank you for your thoughtful review and the time you’ve invested in evaluating our work. We have carefully addressed the points you raised in our rebuttal, and we would greatly appreciate the opportunity to clarify or discuss any remaining questions or concerns you may have.
> >
> > Thank you once again for your valuable feedback.

---

> > > ### Comment · Reviewer_geFN · 2024-08-12
> > > **Response to authors**
> > >
> > > I thank the authors for their detailed response, and apologise for the delay in mine. I appreciate the additional experiments performed using SimCLR and DINO and think these offer valuable extra signal in the evaluation. On the other hand, I was similar to reviewer GyCT confused by the wording of "dynamic acceleration schedule" and thought that this meant it was optimised online. I see now that it is predefined and fixed. Overall, I still see the scope and contribution as the main limitation of this paper, and that this limits its impact.
> > >
> > > Thank you for the high quality responses overall. Having read all reviews and responses, I am happy to stand by my original score of weak accept.

---

> > > > ### Author Response · Authors · 2024-08-13
> > > > **Final comment**
> > > >
> > > > Please read our final message to all reviewers. We truly appreciate your efforts in reviewing our paper and helping us improve it.

---

### Official Review · Reviewer_GyCT · 2024-07-13

**Soundness:** 2
**Presentation:** 3
**Contribution:** 2
**Rating:** 4
**Confidence:** 3

**Summary:**

The paper presents a framework to speed up the pre-training of Vision Transformers (ViTs) in a self-supervised contrastive learning setup. The proposed method incorporates randomized token dropout and flexible patch scaling. The authors leverage this framework to analyze estimated gradient errors and its downstream performance. Additionally, they propose to determine an optimal dynamic acceleration schedule during training. Experimental findings demonstrate improvements in the convergence rate of the MoCo-v3 model across IN-100 and IN-1k datasets.

**Strengths:**

- The proposed acceleration framework brings noticeable improvements in the pre-training convergence of MoCo-v3 on IN-100 and IN-1k.
- The framework incorporates various sequence compression strategies. The authors investigate how these strategies affect gradient estimation errors and analyze their impact on downstream performance.
- The acceleration framework includes a dynamic scheduler that adapts during training. It is validated across different training budgets and supported by ablation studies that highlight the importance of the individual contribution of token dropout and patch-scaling.

**Weaknesses:**

- The proposed framework consists of two components: (1) randomized token dropout and (2) flexible patch scaling. Both ideas exist and have generally been studied for efficient pre-training of ViTs. For instance, the idea of dropping tokens in ViTs has been explored in various forms since their introduction. Recent research has focused on more sophisticated and targeted ways of applying token dropout in ViTs [1, 2, 3]. As a result, the main contributions of this work may be a bit limited.
- In section 4.3 - 'Since there are many more linear operations than quadratic ones, the time complexity of linear operations dominates. Thus, for the sake of simplicity, we consider the time complexity of the ViT architecture to be linear in the sequence length'. This statement seems oversimplified. While it's true that there are more distinct linear operations, this doesn't automatically mean they dominate the time complexity. I am not convinced that having more linear operations would negate the impact of the quadratic operation, even with a sequence length of ~200. This assertion requires further clarification and possibly revision.
- The concepts of randomized token dropout and patch scaling could potentially benefit ViTs in other self-supervised learning (SSL) approaches such as distillation-based SSL (DINO [4], iBOT [5]). I think there will be more contribution if the authors explored pre-training of ViTs in a broader SSL setting than being restricted to just MoCo-v3.
- I am skeptical about labeling the described approach as truly dynamic acceleration. Although it employs cost-adjusted MSE to compare strategies efficiently, the acceleration schedule is predetermined based on the analysis of intermediate checkpoints from a pre-trained model (as mentioned in section 5.1 under Dynamic Acceleration). A genuinely dynamic acceleration approach would involve real-time adjustments during training. Therefore, the method could be better described as an optimized static schedule that varies across different training stages rather than a fully adaptive, dynamic approach.

[1] Marin, Dmitrii, et al. "Token pooling in vision transformers." arXiv preprint arXiv:2110.03860 (2021).

[2] Ryoo, Michael S., et al. "Tokenlearner: What can 8 learned tokens do for images and videos?." arXiv preprint arXiv:2106.11297 (2021).

[3] Wang, Yulin, et al. "Not all images are worth 16x16 words: Dynamic transformers for efficient image recognition." Advances in neural information processing systems 34 (2021): 11960-11973.

[4] Caron, Mathilde, et al. "Emerging properties in self-supervised vision transformers." Proceedings of the IEEE/CVF international conference on computer vision. 2021.

[5] Zhou, Jinghao, et al. "ibot: Image bert pre-training with online tokenizer." arXiv preprint arXiv:2111.07832 (2021).

**Questions:**

- Line 210: 'The variance is a function of their computational cost.' Could you provide references or further clarification on how variance is a direct function of computational cost?
- Line 268 - 'The only modification was the use of a non-symmetric loss.' Is there a specific reason for this, considering that MoCo-v3 uses symmetric loss?
- Line 271 - 'Non-symmetric version produces more diverse batches.' Could you elaborate on what diverse batches mean and how beneficial they are exactly?
- See weaknesses.

**Limitations:**

The limitations are not discussed.

---

> ### Author Rebuttal · Authors · 2024-08-07
>
> We appreciate the valuable feedback. We are glad the reviewer found the noticeable improvements in ViT contrastive pre-training convergence valuable. This is indeed the flagship result of the paper, which has not been explored in any other prior work. Given the importance of the topic (contrastive pretraining is a foundational pretraining technique that consumes considerable amounts of computational resources) and the positive results demonstrated in this paper, we hope the reviewer reconsiders its score for this reason alone. We will do our best to address the raised concerns in the main paper, as we outline below. If there are any remaining concerns we can clarify, or provide additional results/analysis, please let us know. We will be responsive during the author-reviewer discussion phase.
> ## Limited contribution
> While it is true that both token dropout and patch scaling have been used in the literature, these techniques have not been studied for efficient gradient approximation (the main topic of our work). One exception is FLIP [22] where token dropout was used for accelerating vision language pre-training. The reviewer mentions prior work on token pruning and token aggregation techniques [1,2,3], however, these works study inference time acceleration, not acceleration of gradient computation. In our case, while we accelerate training convergence by reducing the compute requirements of each training iteration, we do not change the model architecture or its inference graph in any way. The final pre-trained model is still the original VIT.
>
> More importantly, the paper’s main contribution is the methodology used to obtain the optimized acceleration schedule. The methodology, based on gradient estimation error analysis, is novel and, as shown in the paper. effective. Furthermore, while we focus on contrastive pretraining, the proposed methodology can be generalized to other training regimes. Thus, the publication of this work has the potential for downstream impact beyond contrastive pre-training of VIT models.
>
> ## Approximate linear time complexity of low-resolution VITs
> This statement has been empirically verified. Below we show the time spent on a forward plus backward step for a VIT using varying sequence lengths. Timings were obtained using an RTX A4500 for a batch size of 16. To ensure that pure GPU computation is measured (without any CPU bottlenecks of data loading), randomly generated sequences were used at each iteration. Time measurements were averaged over 20 independent time measurements. As can be seen, the linear approximation is remarkably accurate until a sequence length of 300. In our observations, the quadratic cost of attention mechanisms only becomes a factor at sequence lengths of 750 and above.
>
> | **#Tokens**| 25| 50     | 100    | 150    | 200    | 300    |
> |-|-|-|-|-|-|-|
> | **Time Spent** | 0.030  | 0.046  | 0.086  | 0.132  | 0.171  | 0.267  |
> | **Linear Trendline ($R^2=0.998$)** | 0.024  | 0.047  | 0.089  | 0.133  | 0.176  | 0.263  |
>
> ## Generalization to beyond MoCo
> This is a great suggestion. We have indeed been working on creating optimized acceleration schedules for algorithms beyond MoCo, using the technique proposed in the paper. We were able to achieve a 2.5x speedup in DINO pre-training and a 3.3x speedup in SimCLR pre-training.
>
>
> | Algorithm | Acceleration | Training Budget (M) | NN (%) | LP (%) | FT (%) |
> |---|----|-------|-----|-----|------|
> | **SimCLR** | ✔    | 922      | 50.70        | 68.43    | 81.55    |
> |           | ✗            | 3075        | 50.22        | 68.33    | 81.39    |
> | **DINO (4 small crops)** | ✔      | 1138   | 66.00        | 77.42    | 82.01    |
> |           | ✗            | 2846    | 67.36        | 77.48    | 81.87    |
>
> where NN, LP, and FT refer to the accuracies of near neighbor, linear probing, and fine-tuning, respectively. For SimCLR, we simply replaced the backbone in the original implementation (from a ConvNet to ViT-base). As for DINO, the original method performs contrastive learning on both large and small crops. The small crops have a lower computational burden and thus can have a similar effect of speeding up training. To provide a realistic comparison to DINO, both the baseline (unaccelerated) and our accelerated version still use small crops (4 to be exact) in addition to the 2 large crops, with acceleration only applied to large crops. We will add to the paper the results above as well as an analysis of the impact of varying numbers of small crops in DINO.
>
> ## Naming proposed approach as dynamic vs optimized acceleration
> We agree with the reviewer. “Optimized acceleration schedule” is a more appropriate description of the proposed approach. We will revise the paper accordingly. Thanks for pointing this out.
>
> ## Variance as a function of computational cost
> We clarify what we meant in the next sentence (lines 210-212). In short, assume we can decide how much computational budget to provide to approximate one gradient. A simple strategy to improve this approximation is simply to use bigger batch sizes (ie, averaging the gradient over more samples). While averaging does not change estimation bias, it reduces the estimation variance. That’s why we say that variance is a function of the allocated budget. We will clarify this in the paper.
> ## Non-symmetric loss
> Given constant GPU resources, with a non-symmetric loss we can use twice the batch size. This has two advantages, 1) the number of negatives is doubled (an important parameter in contrastive learning); and 2) sample-wise gradients are obtained from truly independent samples (ie, all gradients are obtained from different images). The second point is what we are referring to when saying that the non-symmetric versions use more diverse batches: ie, instead of using the same image to compute two losses and their respective gradients (which can be correlated), the non-symmetric version simply computes each gradient from completely different images.

---

> > ### Comment · Reviewer_GyCT · 2024-08-10
> > **Response to Rebuttal**
> >
> > Thank you for the responses. Most things have been addressed.
> >
> > I'm not fully convinced by the claim that linear operations dominate the time complexity; when purely speaking asymptotically, we still refer to ViTs as having a quadratic complexity in sequence length. Perhaps consider changing the section header to reflect a focus on low-resolution settings. Overall, I would have preferred a broader focus on general contrastive settings, as promised in the abstract, rather than just MoCo. Also, DINO is not a contrastive setup; it's a distillation-based method. It's great that you're getting good results with it, but I'm unsure if it fits the contrastive theme of the paper.

---

> > > ### Author Response · Authors · 2024-08-10
> > > **Discussion**
> > >
> > > Thank you for engaging during the discussion period.
> > >
> > > **Linear time complexity**
> > >
> > > There's a couple of things we'd wanna add. First, we completely agree that transformers have quadratic time complexity wrt the sequence length. This quadratic complexity is a bottleneck (and thus cannot be ignored) for NLP applications, where sequence length is often in the thousands, and some vision applications like segmentation or image generation where images are processed at high resolution (640x640 and even higher).
> > >
> > > However, contrastive learning (and most other self-supervised learning and recognition applications) has been traditionally studied at the baseline resolution of 224x224 (which with a patch size of 16x16 yields a sequence length of 196). We have not modified the resolution of these algorithms in our paper. So, while we understand the reviewer's point, we used the linear approximation simply because it simplifies how time is accounted for, and because at the standard 224x224 resolution (used in nearly all papers in contrastive learning), the approximation is very accurate. Also, note that the proposed approach would be even more effective if the quadratic operations were the dominant ones, as the impact of reducing the sequence length would be even higher. We will seek to further clarify this in the paper.
> > >
> > > **DINO:**
> > >
> > > While contrastive and distillation-based methods are not exactly the same, they both learn through view invariance (seeking to represent different views by the same embedding). To include DINO and SimCLR results, we would slightly modify the abstract and introduction to be slightly more general (saying "We focus our evaluation efforts on view-invariance pretraining methods including contrastive learning algorithms like MoCo and SimCLR and distillation-based methods like DINO."). Please let us know if you think this would still not be appropriate.

---

> > > > ### Author Response · Authors · 2024-08-13
> > > > **Final comment**
> > > >
> > > > Dear reviewer, please read our final message to all reviewers. We truly appreciate your efforts in reviewing our paper. We hope you consider upgrading the final score to reflect the significant improvements made to the paper during the review cycle (many of them as a result of your valuable suggestions).

---

### Author Response · Authors · 2024-08-13
**Post-rebuttal appeal**

We thank again all reviewers for the thoughtful comments, and for helping us improve our paper. As a result of this review process, we have **further validated the proposed acceleration framework with two other self-supervised methods** (DINO and SimCLR) and strengthened each comparison by conducting **finetuning experiments**. We will also revise the paper to clarify our contributions, as we’ve detailed before.

Overall, our paper proposes a pretraining acceleration method that is shown to **significantly speed up the convergence of a variety of self-supervised methods**, including constrastive frameworks (MoCo, SimCLR), and distillation-based frameworks (DINO) **demonstrating its broad usability**. Given the generality of the proposed method (as acknowledged by several reviewers), we believe it also has the potential to accelerate pre-training frameworks beyond those investigated in the paper.

Finally, given the compute-intensive nature of model pre-training and its implications on reproducibility, energy costs, and carbon footprints, we believe that **further research on accelerated training is essential for advancing sustainable AI practices. Our paper aims to inspire continued exploration in this area**, promoting the development of more efficient training methodologies that can 1) reduce the environmental impact of machine learning and 2) improve accessibility for researchers with limited resources. In light of these considerations and the significant improvements made to the paper during the review cycle, we ask all reviewers to reassess their final scores.

Best regards,

The authors

---

### Decision · Program_Chairs · 2024-09-25

**Decision:**

Accept (poster)

**Comment:**

This work explores two approaches to accelerate contrastive learning with vision transformers (ViTs) and improve pretraining efficiency. The authors also analyze gradient estimation errors related to these methods and propose an automated strategy for optimal acceleration during pretraining. The approach achieves comparable performance. In the rebuttal, the authors provided additional experiments addressing the reviewers' feedback. We encourage the authors to fully incorporate the reviewers' comments, clarify the major contributions of the proposed solution, and resolve any inconsistencies, such as whether Dynamic Acceleration is applied on-the-fly or as post-processing.